# Pharmacological Screening of Venoms from Five Brazilian *Micrurus* Species on Different Ion Channels

**DOI:** 10.3390/ijms23147714

**Published:** 2022-07-13

**Authors:** Jessica Matos Kleiz-Ferreira, Hans Bernaerts, Ernesto Lopes Pinheiro-Junior, Steve Peigneur, Russolina Benedeta Zingali, Jan Tytgat

**Affiliations:** 1Laboratório de Hemostase e Veneno, Instituto de Bioquímica Médica Leopoldo de Meis (IBqM) and Instituto Nacional de Ciência e Tecnologia de Biologia Estrutural e Bioimagem (Inbeb), Universidade Federal do Rio de Janeiro (UFRJ), Rio de Janeiro 21941-902, Brazil; jessica.kleiz@bioqmed.ufrj.br; 2Toxicology and Pharmacology, Campus Gasthuisberg O&N2, University of Leuven (KU Leuven), Herestraat 49, P.O. Box 922, B-3000 Leuven, Belgium; hans.bernaerts1@hotmail.com (H.B.); ernesto.lopes@kuleuven.be (E.L.P.-J.); steve.peigneur@kuleuven.be (S.P.); jan.tytgat@kuleuven.be (J.T.)

**Keywords:** *Micrurus* venoms, neurotoxins, two-microelectrode voltage clamp, ion channels

## Abstract

Coral snake venoms from the *Micrurus* genus are a natural library of components with multiple targets, yet are poorly explored. In Brazil, 34 *Micrurus* species are currently described, and just a few have been investigated for their venom activities. *Micrurus* venoms are composed mainly of phospholipases A_2_ and three-finger toxins, which are responsible for neuromuscular blockade—the main envenomation outcome in humans. Beyond these two major toxin families, minor components are also important for the global venom activity, including Kunitz-peptides, serine proteases, 5′ nucleotidases, among others. In the present study, we used the two-microelectrode voltage clamp technique to explore the crude venom activities of five different *Micrurus* species from the south and southeast of Brazil: *M. altirostris*, *M. corallinus*, *M. frontalis*, *M. carvalhoi* and *M. decoratus*. All five venoms induced full inhibition of the muscle-type α1β1δε nAChR with different levels of reversibility. We found *M. altirostris* and *M. frontalis* venoms acting as partial inhibitors of the neuronal-type α7 nAChR with an interesting subsequent potentiation after one washout. We discovered that *M. altirostris* and *M. corallinus* venoms modulate the α1β2 GABA_A_R. Interestingly, the screening on K_V_1.3 showed that all five *Micrurus* venoms act as inhibitors, being totally reversible after the washout. Since this activity seems to be conserved among different species, we hypothesized that the *Micrurus* venoms may rely on potassium channel inhibitory activity as an important feature of their envenomation strategy. Finally, tests on Na_V_1.2 and Na_V_1.4 showed that these channels do not seem to be targeted by *Micrurus* venoms. In summary, the venoms tested are multifunctional, each of them acting on at least two different types of targets.

## 1. Introduction

In Brazil, most snake species from the Elapid family are from the *Micrurus* genus, popularly known as coral snakes. To date, 34 *Micrurus* species found throughout the country in diverse biomes are described [1]. *Micrurus* venoms are composed mainly of two toxin families, the phospholipase A_2_ (PLA_2_) [2] and the three-finger toxin (3FTx) [3,4], conferring a phenotypic dichotomy pattern [5]. This dichotomy has been better correlated to the geographic distribution of *Micrurus* species among the north-south axis of the American continent than to the species phylogeny [6]. The further to the north of the continent, more PLA_2_-predominant venoms are found, and the further to the south of the continent, more 3FTx-predominant venoms are found [6,7]. Moreover, other minor or less frequent venom components were found in transcriptomes and proteomes analyses of *Micrurus* venom glands and venoms, such as L-amino acid oxidase (LAO), snake venom metalloproteinase (SVMP), snake venom serine proteinase (SVSP), Kunitz serine protease inhibitors (KSPI), 5′ nucleotidase (5′NUC), aminopeptidase, phosphodiesterase (PDE), hyaluronidase (HYAL), dipeptidyl peptidases (DPP), cysteine-rich secretory protein (CRISP), C-type lectin (CTL), lipases, vascular endothelial growth factor (VEGF), nerve growth factor (NGF), phospholipase B (PLB), wraprin-like proteins, veficolins (VEF) and vespryns (VESP) [5,8,9,10,11,12,13,14,15]. In Brazil, *Micrurus* venoms are also predominant in 3FTx and PLA_2_, with the majority being 3FTx-predominant, in accordance with the dichotomic pattern [5,7,8,9,10]. Studies on venom proteomes of Brazilian *Micrurus* species have shown the variety of different sequences of both 3FTx and PLA_2_ proteins that can be found in one single venom [8,9]. For instance, in the *M. frontalis* venom proteome, 41 different types of 3FTx sequences and 19 types of PLA_2_s have been found [10]. This protein diversity probably leads to different biological activities that contribute to the envenomation outcome. We know for the 3FTxs, for example, that we can find a range of toxins with different functions depending on the protein sequence and local structures [3,16]. Importantly, despite the diversified toxins that we can find in Brazilian *Micrurus* species [8,16], currently few proteins were explored and characterized in terms of their structures and functions [17,18,19,20].

Based on the *Micrurus* venom profile, the clinical outcome of envenomation and laboratory experiments, coral snake venoms are considered neurotoxic, myotoxic, cardiotoxic and hemorrhagic [21,22,23]. The neuromuscular blockade is the hallmark of the *Micrurus* envenomation [14,21,22,24,25,26] and the two most abundant neurotoxin families are the main ones responsible for this outcome. Diverse types of presynaptically active PLA_2_ can act by blocking the release of acetylcholine (ACh) [27,28,29,30], and different types of postsynaptic 3FTxs can act by blocking the nicotinic acetylcholine receptors (nAChRs) [31], leading to neuromuscular paralysis. Beyond the classical activities, the exploration of 3FTxs and PLA_2_s from diverse Elapids around the globe (e.g., coral snakes, Najas, Bungarus, Mambas, etc.) has revealed a wide range of possible targets for these two variables and versatile groups of toxins, especially the 3FTxs [32]. For instance, 3FTxs have been reported acting on the GABA_A_ Receptor (GABA_A_R) [33,34,35], adrenoreceptors [36,37,38], potassium channel [39], acid-sensing ion channel (ASIC) [40] and others, as recently reviewed [32].

In Brazil, the comprehension of the *Micrurus* venom activities, especially of isolated toxins, is still in progress. Considering the diverse number of species that we already have described in Brazil, we are just at the very tip of the exploration of their toxins, their pharmacological properties, and possible biotechnological and therapeutical applications.

In the present work, we investigated the crude venoms from five different *Micrurus* species from the south and southeast regions of Brazil: *Micrurus corallinus*, *M. frontalis*, *M. carvalhoi* (former *M. lemniscatus carvalhoi*) and *M. decoratus* from the southeast, and *M. altirostris* from the south of Brazil (Figure 1). To explore the *Micrurus* venoms’ functions, we injected cRNAs of different types of ion channels in *Xenopus laevis* oocytes and measured the whole-cell current in the presence of the venoms by using an electrophysiology assay, named the two-microelectrode voltage clamp (TEVC) technique. In this way, we were able to screen the venoms on different targets, obtaining an overview of their activities. This screening is important to understand other possible functions for *Micrurus* venoms beyond the ones already reported in the literature, as well as to guide the search for isolated molecules with specific targets of pharmacological interest.

## 2. Results

### 2.1. Chromatographic Profile of Five Micrurus Venoms from the South and Southeast Brazil

In the literature, venom proteomics analyses for *M. altirostris*, *M. corallinus*, *M. frontalis* and *M. carvalhoi* (the old subspecies *M. lemniscatus carvalhoi*) species [7,9,10] are available. For *M. decoratus* venom, there is no proteomic information available that is up to date. To better compare the venoms and their activities, and to understand the differences and similarities among their general protein profile, we fractionated the crude venoms of all the five species studied by RP-HPLC. To the best of our knowledge, this is the first report in the literature of a RP-HPLC profile for *M. decoratus* venom. We can observe that the most complex chromatographic profiles, in terms of the number of fractions, are found for *M. altirostris*, *M. frontalis* and *M. carvalhoi* venoms, ranging from 29 to 35 fractions. On the other hand, *M. corallinus* and *M. decoratus* venom chromatograms contain approximately 15 fractions (Figure 2).

### 2.2. Micrurus Venoms Are Not Cytotoxic to X. laevis Oocytes

Venoms are complex mixtures of toxins and are designed to cause a variety of damages to many different organisms. One of the well reported *Micrurus* venom activities is myotoxicity. In this way, as a first step, we decided to test whether these venoms can cause cytotoxicity to *X. laevis* oocyte, specifically. We used the TEVC technique to measure the electric current in non-injected oocytes after the incubation with *Micrurus* crude venoms, using a ramp protocol, and compared it with the control current before the exposure to venom. The amount of 50 µg of *M. altirostris*, *M. corallinus*, *M. frontalis* and *M. carvalhoi* crude venoms did not cause any cytotoxic effect on oocytes (Figure 3A), while 4 µg of *Apis mellifera* crude venom, used as a positive control, was enough to cause a prominent cytotoxic effect (Figure 3B). *A. mellifera* venom is already known in the literature to cause cytotoxic/cytolytic effects in different cells [41,42,43,44,45,46]. Interestingly, for the *M. frontalis* venom, we found an outward current around five times bigger when compared to the control (Figure 3A,C). It means that there is/are molecule(s) in this venom that is/are causing a specific efflux of ions that is not related to a non-selective pore-forming effect in the cell membrane, as we did not find a change in reversal potential when comparing to the control situation (Figure 3D).

Since the four venoms tested did not cause cytotoxic/cytolytic damage to oocytes, we proceeded to the exploration of their activities in oocytes expressing different types of ligand-gated and voltage-gated ion channels (Section 2.4). Because of the limitation in obtaining *M. decoratus* venom, we did not test the cytotoxic effect in non-injected cells.

### 2.3. Micrurus Frontalis Venom Evoked Cl^−^ Outward Current by Acting Directly or Indirectly on Ca^2+^-Activated Cl^−^ Channels

In the previously cytotoxic experiment with non-injected oocytes, we found an outward current evoked by *M. frontalis* venom, not observed in any other *Micrurus* venom tested (Figure 3A,C). We hypothesized that the outward current observed was Cl^-^ ions flowing throughout the cell by direct or indirect activation of Ca^2+^-activated Cl^-^ channels. To address our hypothesis, we blocked this channel by using niflumic acid (NA), a non-steroidal anti-inflammatory agent that blocks Ca^2+^-activated Cl^−^ channels [47], and evaluated the venom response. In the presence of 100 µM of NA with subsequent incubation of 50 µg of *M. frontalis* venom, we could not observe the venom activity any longer. It means that in the presence of a chloride channel blocker, the venom cannot act, causing the outward current observed previously, suggesting that the *M. frontalis* venom may be interfering with either (1) an increase in the intracellular Ca^2+^ (from external or internal stores), or (2) by directly acting on Ca^2+^-activated Cl^−^ channels (Figure 4).

### 2.4. Micrurus Venoms Act in Different Ion Channels

In order to explore the venoms’ activities on different ligand-gated and voltage-gated ion channels, we expressed different targets in *X. laevis* oocytes and measured the venom activity using the TEVC. We tested three different ligand-gated ion channels: the muscle-type α1β1δε nicotinic acetylcholine receptor (nAChR), neuronal-type α7 nAChR and α1β2 GABA_A_R. Briefly, we first applied the agonist (acetylcholine-ACh or γ-aminobutyric acid-GABA) under the flow to obtain controls, then incubated the cell with 10 µg of crude venom under no flow, and subsequently applied the agonist to check the venom activity. Moreover, washout and ACh pulses were given to check whether and how the activities observed would return to the control stage or not. The five Micrurus venoms showed full inhibition of the muscle-type α1β1δε nAChR (Figure 5A). The cell washout followed by agonist pulses application revealed that each venom has different activities and affinities to the receptor. *M. altirostris*, *M. frontalis* and *M. corallinus* have a clear reversible activity, with the *M. corallinus* venom being almost completely reversible after three washout runs. For *M. altirostris* and *M. frontalis*, a prolonged washout had to be carried out to return to the control level. We found *M. carvalhoi* and *M. decoratus* venoms to be the most potent and almost irreversible in blocking the α1β1δε nAChR (Figure 5A).

Different from the muscle-type nAChR, the α7 neuronal-type was the target for just two venoms, *M. altirostris* and *M. frontalis*. Interestingly, the activity of these two venoms on α7 nAChR was first a partial inhibition and subsequently, after one washout, a potentiation of the current in relation to the control. It means that there are different toxins in the same venom acting differently, as a blocker and as an enhancer, and also with different affinity to the receptor (Figure 5B). For *M. corallinus* and *M.*
*carvalhoi*, no significant activity was observed. *M. decoratus* venom was not included in the experiment with α7 nAChR due to the limited amount of venom available.

For α1β2 GABA_A_R, we found that *M. altirostris* and *M. corallinus* venoms could partially block and modulate the activity of this receptor. Intriguingly, after the venom incubation, we could observe an inversion of the activation and the desensitization behavior of the receptor. In a control condition, the receptor activates quickly until it starts slowly desensitizing under GABA agonist stimulus. On the contrary, after the venom incubation, we could observe a slower activation of the receptor together with an inhibition, and faster desensitization. The mechanism underlying these venoms’ activity must be further explored to understand how the toxins are acting on the inversion of the activation and desensitization processes of the GABA_A_R. For M. frontalis and *M.*
*carvalhoi* venoms, no activity was observed on this receptor (Figure 5C). A preliminary study with *M. decoratus* venom on α1β2 GABA_A_R showed no effect of this venom on this receptor (data not shown).

Following the screening of *Micrurus* venom activities on different ligand-gated ion channels, we evaluated their activities on voltage-gated potassium (K_V_) and sodium (Na_V_) channels: the Shaker-like K_V_1.3, and the TTX-sensitive Na_V_1.2 and Na_V_1.4. All the five venoms showed inhibition on K_V_1.3. *M. corallinus* was the most potent venom in blocking the channel current (48.7% ± 0.1), followed by *M.*
*carvalhoi* (42.6% ± 1) and *M. altirostris* (32.8% ± 1.9) venoms. *M. frontalis* and *M. decoratus* venoms showed less activity compared to the other venoms (respectively, 28.9% ± 2.2 and 23.3% ± 0.8) (Figure 6A). Furthermore, the activity of all the five venoms was completely reversible after the washout (Figure 6A). Experiments with Na_V_1.2 (Figure 6B) showed that the five *Micrurus* venoms have no effect on these channels. Tests on Na_V_1.4 also demonstrated no effect of the five venoms on this subtype (data not shown).

## 3. Discussion

The proteomic and transcriptomic analyses of the *Micrurus* venoms are key contributors to understand the amount, distribution and diversity of the major toxins and other components in these venoms. Based on the venom proteomes available in the literature for the *M. altirostris*, *M. corallinus* and *M. lemniscatus* (=*M. carvalhoi*), we know that there is a predominance of 3FTxs, which stands for 79.5%, 81.7% and 76.7% in these venoms, respectively [7,9,10]. In contrast, *M. frontalis* venom has more PLA_2_ (49.2%) than 3FTx (42.4%) [19]. According to these proteomes [7,9,10] and the chromatographic profiles for the five *Micrurus* species venoms shown in this work (Figure 2A–E) [7,9,10], we can consider three main toxin distribution regions in the chromatogram. The beginning of the chromatogram is rich in 3FTxs, followed by the middle which is rich in PLA_2_s, and finally the end of the chromatogram where the toxins with higher molecular weight are found, such as SVMP, LAO and lipases [7,9,10]. The *M. altirostris*, *M. corallinus*, *M. frontalis* and *M. carvalhoi* RP-HPLC venom profiles reported in this work corroborate the profiles reported in the literature with few differences, which are normally found in different pools of venoms from different animals (Figure 2A–E). The *M. decoratus* RP-HPLC venom profile (Figure 2E), analyzed for the first time in this study, is considerably less complex in fractions when compared to the *M. frontalis* venom (Figure 2C), for example. Comparing other *Micrurus* chromatograms with the venom proteomes described, we can suggest that at least the first two fractions of the *M. decoratus* venom are significantly composed of 3FTxs (Figure 2E). In addition, we hypothesized that the low complexity of the fractions in this venom may suggest that it is 3FTx-predominant with a low amount and diversity of PLA_2_ toxins. The *M. decoratus* venom proteome is under investigation.

Diverse Viperids and Elapid snake venoms are known to be cytotoxic [48,49]. Venoms from Najas are good examples of well-described cytotoxic activity in the Elapid family [50,51,52,53]. In the case of *Micrurus* venoms, many species were reported as being myotoxic, causing muscle cell death [54,55,56,57]. For example, the isolated Lemnitoxin, a member of the PLA_2_ protein family and a major component in *M. lemniscatus* venom, was described as having myotoxic and pro-inflammatory activity [58].

Despite the myotoxicity activity found in several *Micrurus* species, it is not the front-line activity of these venoms to subdue the preys. Nevertheless, to test the activity of the venom in *X. laevis* oocytes without masking specific activities or causing cell death, we first checked whether they were cytotoxic for these cells. All the four venoms tested were not cytotoxic after the exposure of the oocytes using a saturating amount of venom, as shown in Figure 3A. In contrast, we used, as a positive control, the venom from the *Apis mellifera* bee species, well known as having the activity of causing non-ion-selective pores in cell membranes, a cytotoxic/cytolytic effect [41,42,43,44,45,46]. In the electrophysiology experiments using the voltage ramp protocol, when a given venom is forming pores that are non-selective for ions and their conduction, typically two phenomena are observed: (1) a shift of the reversal potential to (wards) 0 mV, and (2) a significant increase in the chord conductance. These two aspects can be clearly seen in the case of the bee venom (Figure 3B,D). We indeed could observe for the bee venom that the voltage at which the potential reverses is situated near 0 mV (slightly positive~3.4 mV), while for *Micrurus* venoms, it remains negative. An offshoot of the cytotoxic experiments was the activity found in the *M. frontalis* venom; here, a unique activity was observed by the evocation of an outward current observed in a time-dependent manner (Figure 3A). When blocking the (presumed) Ca^2+^-activated Cl^−^ channels using the blocker NA, we no longer observed the outward current caused by the *M. frontalis* venom. This is suggestive of the fact that in the presence of the *M. frontalis* venom, an increase in the intracellular calcium is taking place. Components of *M. frontalis* venom may be acting directly on this channel (a Ca^2+^-activated Cl^−^ channel) or most likely acting on the cell membrane, causing this effect indirectly, possibly via the action of presynaptic PLA_2_s.

*Micrurus* venoms present a preponderant neurotoxic activity, and the main outcome of the envenomation is the blockade of the neuromuscular junction at the peripheral nervous system [21,23]. The action of pre- and postsynaptic neurotoxins, generally PLA_2_s (also known as β-neurotoxins) and the 3FTxs (or α-neurotoxins), respectively, can act as the main players to the final venom activity. In general, β-neurotoxins can alter the release of acetylcholine, while α-neurotoxins can bind and modulate the cholinergic receptors. Despite this, other mechanisms of action for these α- and β-neurotoxins can be found, and some are still obscure [31,59,60,61]. Furthermore, a diverse range of activities is described for the 3FTx family [32].

Besides the major toxins, the neurotoxicity of *Micrurus* venoms is complex and other minor molecules can act synergistically, being also important for the envenomation strategy. Based on the classical and better-described activity for the coral snake venoms—the action on cholinergic receptors—in the present work, we tested the five venoms on muscle and neuronal types of nAChR and on GABA_A_R. The challenge of these venoms against the muscle type α1β1δε nAChR shows that all of them can fully block the receptor, though interestingly differing in the response after the washing of the venom and application of the subsequent pulses of acetylcholine. We can point out two activities found in opposite extremes: *M. corallinus* venom is almost completely reversible after the third pulse of ACh, while the *M. decoratus* venom demonstrated no significant reversibility until the sixth pulse (Figure 5A).

All the five venoms showed different reversibility, which is compatible with the diversity of the venom profiles that is leading to different responses. Vital Brazil in 1987, in experiments with isolated rat phrenic nerve-diaphragm, observed that the *M. frontalis* venom from São Paulo state (Brazil) was reversible for the neuromuscular blockade, while the *M. frontalis* venom from Mato Grosso (Brazil) was interestingly irreversible [23]. This experiment illustrates that even within the same species, we can find variation in the reversibility, which, in this case, is related to the geographic distribution of the animal. Moreover, pre- and postsynaptic actions can generate different outcomes for the same receptor. Finally, the full blockage of the muscle-type nAChR by the five different *Micrurus* venoms reinforces the conserved and potent activity of these venoms on cholinergic receptors. Despite the wide range of biological targets, classically, the main 3FTx activity is the disturbance of the cholinergic system, specifically the inhibition of the muscarinic or nicotinic AChRs [31].

The neuronal-type α7 nAChR can also be a target for *Micrurus* venom toxins. An example is the α-neurotoxin from the venom of the Mexican coral snake *M. laticollaris*, which blocks not only the muscle-type, but also the α7 neuronal-type of nAChR [62]. In the case of the five *Micrurus* venoms tested on α7 nAChR in the present work, the most interesting activity was the potentiation of the receptor after the partial blockage caused by the *M. altirostris* and *M. frontalis* venoms. At this point, we can only speculate that different toxins with different kinetics for this receptor are taking part in this observation.

The α-neurotoxins can be multifunctional and bind not just to the cholinergic receptors but also to other receptors, such as the GABA_A_R [35,63]. For example, findings in studies with α-bungarotoxin showed that besides the classical activity reported on the nAChR, this protein can block functional and non-functional GABA_A_R [34,64]. In *Micrurus*, two toxins of the 3FTx family from the venom of the Costa Rican coral snake *M. mipartitus*, named MmTX1 and MmTX2, were reported as strong modulators of GABA_A_R by allosterically increasing the receptor sensitivity to the agonist [33]. Among the five *Micrurus* venoms tested here, two showed an interesting modulation of GABA_A_R.: *M. altirostris* and *M. corallinus* venoms, besides partially blocking the receptor, curiously inverted the speed for the activation and desensitization of the receptor when compared to the control. To the best of our knowledge, this modulation was not seen yet for snake toxins and deserves further investigation. Moreover, we decided to use the α1β2 GABA_A_R subunits, since previous and preliminary tests using α1β2γ2 subunits showed that the α and β were important for the activity while the γ was dispensable (data not shown). This result corroborates with Jean-Pierre Rosso et al. experiments, where two toxins from *M. mipartitus* venom were tested using the same subunits and it was verified that the γ subunit was not required for the activity [33].

Voltage-gated ion channels are also targets for many venom-derived peptides and proteins [65,66,67,68]. These channels are widely distributed in the human body and are critical in several physiological processes [69] and consequently are involved in many diseases. For example, voltage-gated potassium channels (K_V_s) [70,71] and voltage-gated sodium channels (Na_V_s) [70,72] are objects of study in high incidence diseases such as cancer and neurodegenerative disorders [73,74,75,76], and so, are explored as therapeutical targets [77,78,79,80]. Toxins from sea anemone, scorpions, spiders, cone snails, centipedes and snakes are being investigated in studies with K_V_ and Na_V_ [65,67,81,82,83,84,85,86,87,88]. Therefore, in general, animal toxins are of pharmacological interest in the studies using voltage-gated ion channels.

Since *Micrurus* venoms have been proved to be versatile and often promiscuous in terms of activity and selectivity, as shown in this work, we decided to check whether these venoms can act on voltage-gated ion channels. We tested the *Micrurus* venoms on K_V_1.3, noticeably a promising channel for therapeutic drugs [79,89,90] and a known target for toxins [74]. Literature evidence indeed proposes K_V_1.3 as a key target for multi-therapeutic applications in many human diseases. This channel is mainly found in the immune system and nerve cells, and it can be involved in inflammatory processes, autoimmune disorders, cancer, obesity, etc. [91,92,93]. Specifically, K_V_1.3 inhibitory molecules are of pharmacological interest since they are considered suppressors for cell proliferation [94,95]. It is worth mentioning that peptides from scorpion venoms are largely found as inhibitors of potassium channels, including the K_V_1.3 [96,97,98,99]. Perhaps, the natural source with the highest number of peptide modulators of potassium channels discovered to date comes from scorpion venoms [96,98].

In the present work, we demonstrated that the five *Micrurus* crude venoms tested on K_V_1.3 can inhibit this channel (Figure 6A), with all of them being reversible after the washout. Notably, an isolated PLA_2_ toxin called MiDCA1, from *Micrurus dumerilii carinicauda*, a species from the north of South America, showed reversible inhibition of the K_V_2.1 [100]. According to our research, only this isolated toxin was reported in *Micrurus* venoms as a potassium channel inhibitor, to date [100]. This finding, together with our results, led us to hypothesize that this activity may be evolutionarily conserved among this genus and that it might be important for the toxin arsenal to subdue prey. Since we found activity on potassium channels for all the crude venoms tested, we speculate that *Micrurus* venoms are also a rich source for potassium channel modulators still to be explored.

Finally, we tested the five venoms on Na_V_1.2 (Figure 6B) and Na_V_1.4; none of them have demonstrated activity on either channel subtypes. Contrary to the potassium channels, this result shows that sodium channels do not seem to be a target for *Micrurus* venoms, at least for the five coral snake venoms tested from south and southeast of Brazil on these channel subtypes. In general, sodium channels have not been systematically investigated for coral snake venoms around the world. Nevertheless, an Elapidae species from Indonesia called Long-glanded blue coral snake (*Calliophis bivirgatus*) was reported as “a snake with the scorpion’s sting”, due to a toxin from the 3FTx family found in its venom which activates the Na_V_1.4 [101]. Additional investigations of *Micrurus* venoms are needed for a better understanding of whether or not sodium channels represent a target.

## 4. Conclusions

In this work, we explored the venom activity of five different *Micrurus* species from the south and southeast Brazil on six different types of ion channels: three ligand-gated ion channels and three voltage-gated ion channels. We discovered that these venoms can be multifunctional, each of them acting on at least two different types of targets.

Taking into consideration the expected activity on the muscle-type of nAChR, for *Micrurus* venoms, we demonstrated that in fact all of the five venoms were capable of fully blocking this channel but differed in their reversibility. This result shows that besides the conserved activity, each venom has its particularities. Corroborating with this idea, just some venoms demonstrated activity on GABA_A_R and the neuronal-type of nAChR. Moreover, we showed that *Micrurus* venoms may rely on potassium channel inhibitory activity as an important feature of their envenomation strategy since it seems to be conserved among the genus. On the other hand, studies with other species must be carried out to verify if this is indeed a hallmark of the *Micrurus* genus. Further studies on the specific toxins responsible for the functional effects described in this paper are now ongoing. In conclusion, *Micrurus* venoms demonstrate to be an invaluable pharmacological and biotechnological source, and so, are noteworthy of deeper investigation.

## 5. Materials and Methods

### 5.1. Venoms

Five venoms from different *Micrurus* species were used in this work: *Micrurus altirostris*, *Micrurus corallinus*, *Micrurus frontalis*, *Micrurus*
*carvalhoi* (former *M. lemniscatus carvalhoi*) and *Micrurus decoratus* (Sisgen code A9F3694). The venoms were extracted from specimens kept in serpentarium and kindly given by the Núcleo de Ofiologia de Porto Alegre-NOPA (*M. altirostris*) and Instituto Vital Brazil (*M. corallinus*, *M. frontalis*, *M.*
*carvalhoi* and *M. decoratus*). The venoms were centrifuged to remove debris, lyophilized and stored at −20 to be used further. For all the experiments, pools of adult venoms were used with the exception of *M. decoratus* venom, which just one specimen was used for the venom extraction. *M. decoratus* is a rare species and difficult to obtain, being a limiting factor to study its venom.

### 5.2. RP-HPLC Profile

The crude venom chromatographic profiles were obtained using a reverse-phase high performance liquid chromatography (RP-HPLC). Approximately 500 µg of lyophilized venom was resuspended in 500 µL of 5% acetonitrile and 0.1% trifluoroacetic acid (TFA) solution and centrifuged at 8000× *g* for 10 min to remove the remaining debris. We used a Teknokroma Europa C18 column (0.4 cm × 25 cm, 5 mm particle size, 300 Å pore size) in a high-performance liquid chromatographic system (HPLC-Shimadzu). The flow rate was 1 mL/min and the experiments were carried out in a linear gradient of 0.1% TFA in water (solution A) and 0.1% TFA in acetonitrile (solution B) in the following condition: 5 min of isocratically 5% of B solution; from 5 to 85 min, 5 to 70% of B solution; 85 to 90 min, 70% of B solution; 90 to 95 min, 5% of B solution. The column was equilibrated with 5% of B solution. The protein detection was monitored at 214 and 280 nm (λ) and the graphs were generated in GraphPad Prism, version 8.0.2 (GraphPad Software, San Diego, CA, USA).

### 5.3. Electrophysiologic Recordings on Xenopus laevis Oocytes

In order to measure the activity of the venoms in non-injected and injected oocytes with cRNA encoding ion channels, we used the two-electrode voltage clamp (TEVC) technique. The electrophysiologic recordings were carried out on a GeneClamp 500B amplifier equipment (Axon Instruments^®^, Foster City, CA, USA) controlled by Clampex 10.4 or pCLAMP 10.1 software (Molecular Device, San José, CA, USA) for the data acquisition. Two micropipettes were prepared using capillary glasses (borosilicate WPI 1B120-6) in a microelectrode puller (PUL-1, World Precision Instruments^®^, Sarasota, FL, USA). Voltage and current electrodes were filled with 3 M KCl and the resistances of both electrodes were kept between 1.0 to 1.5 MΩ. Oocytes were placed in a 0.2 mL recording chamber with a perfusing system. For the whole-cell current recordings, the leak subtraction was carried out with P/4 protocol. The experiments were carried out at room temperature (18 to 22 °C). The data were analyzed in the ClampFit software version 11.0.3 (Molecular Devices, San José, CA, USA) and the final graphs were generated using GraphPad Prism, version 8.0.2 (GraphPad Software, San Diego, CA, USA).

*Xenopus laevis* frogs used to extract the oocytes were purchased from Nasco (Fort Atkinson, WI, USA) and maintained in the Aquatic Facility of KU Leuven, according to the European Union regulation concerning the welfare of laboratory animals established in the directive 2010/63/EU. The use of *X. laevis* oocytes was approved by the Animal Ethics Committee of KU Leuven licensed by the number P186/2019. The oocytes in stages V–VI were surgically extracted from mature female animals and anesthetized with 0.1% tricaine solution (aminobenzoic acid ethyl ester, Merck^®^, Darmstadt, Germany). The oocyte vitellin membranes were defolliculated by incubating with 2.5 mg/mL of collagenase in Ca^2+^ free ND96 solution (96 mM NaCl; 2 mM KCl; 2 mM MgCl_2_ and 5 mM HEPES, pH 7.5) at 16 °C, for 1 h 45 min. The defolliculated oocytes were maintained in ND96 (96 mM NaCl; 2 mM KCl; 1.8 mM CaCl_2_; 2 mM MgCl_2_ and 5 mM HEPES, pH 7.5) solution supplemented with Theofylline and gentamicin at 16 °C to be used further.

#### 5.3.1. Cytotoxic Experiments with Non-Injected Oocytes

In order to check whether the venoms were cytotoxic to oocytes or not, we used non-injected oocytes, without expressing any particular type of ion channel. For the electrophysiologic measurements, we used a 2-s ramp protocol from −120 to 70 mV, with cells clamped at −20 mV holding potential. Lyophilized venoms were resuspended in ND96 solution for the final concentration of ~0.5 to 10 mg/mL. An amount of 50 µg of crude venom was applied by pipetting directly on the bath, under no flow. The bath perfusion of ND96 solution to wash the oocytes was under the flow (~2 mL/min).

#### 5.3.2. Niflumic Acid Assay

For the electrophysiologic recordings, the ramp protocol was used as described above. In order to verify if outward current after *M. frontalis* venom application was evoked due to the activation of Ca^2+^-activated Cl^−^ channels, after 20 sweeps of 2-s controls, we blocked Ca^2+^-activated Cl^−^ channels by applying the niflumic acid reagent (Sigma^®^, St. Louis, MO, USA) directly in the bath by pipetting, for a final concentration of 100 µM. Subsequently, 50 µg of *M. frontalis* crude venom was applied also by pipetting directly on the bath and the response was evaluated.

#### 5.3.3. Ion Channels Heterologous Expression and Recordings in Oocytes

Six different types of ion channels were used in the present work: (1) human muscle-type of α1β1δε nAChR; (2) human neuronal-type of α7 nAChR; (3) human α1β2 GABA_A_R; (4) human voltage-gated potassium channel (K_V_1.3); and (5, 6) rat voltage-gated sodium channels (Na_V_1.2 and Na_V_1.4), coexpressed with their rat auxiliary β subunits. For the heterologous expression in oocytes, plasmids for each channel subunit were linearized using appropriate restriction enzymes and transcribed using a T7 or SP6 mMESSAGEmMACHINE transcription kit (Ambion, Austin, TX, USA). For the channel expression, the oocytes were microinjected with cRNA using a micro-injector (Drummond Scientifc^®^, Broomall, PA, USA) with a final volume ranging from 4 to 60 nL per cell, depending on the channel subtype and the cRNA concentration. For the injection of α1β2 GABA_A_R and muscle-type α1β1δε nAChR, we used the proportion of 1:1 and 2:1:1:1 of the subunits, respectively. For the injection of both Na_V_ channel types, the proportion of 1:1 was used for the channel and the auxiliary β subunit. The oocytes were incubated in ND96 solution supplemented with 50 mg/L of gentamicin sulfate and 180 mg/L of Theophylline, and kept at 16 °C for 1 to 5 days to express the channels. For the experiments with the ligand-gated ion channels (muscle-type α1β1δε nAChR, neuronal-type α7 nAChR and α1β2 GABA_A_R), the activation by ligand agonists was also carried out under gravitational flow. In the case of voltage-gated channels (K_V_1.3, Na_V_1.2 and Na_V_1.4), protocols using different voltage steps, depending on the isoform studied, were applied.

For the whole-cell current recordings, different protocols were used depending on the ion channel measured, as follows: For muscle-type α1β1δε nAChR, cells were clamped at −70 mV holding potential. A protocol of 3 pulses of 100 µM acetylcholine (Sigma^®^, St. Louis, MO, USA) agonist with an interval of ~30 s between each pulse was applied to make controls for each cell experiment. The cell washout was carried out before venom application and ACh pulses were applied to test the venom activity, as in the control. For neuronal-type α7 nAChR, cells were also clamped at −70 mV and 2 or 3 100 µM ACh pulses were applied as the control, followed by cell washout, venom application and ACh pulses to test the activity. For α1β2 GABA_A_R, cells were clamped at −90 mV. A protocol of 2 pulses of 100 mM of γ-aminobutyric acid (GABA) agonist (Sigma^®^, St. Louis, MO, USA) for ~30 s and ~3 min of a washout interval between each pulse was performed. Cells were also washed after the controls, to apply the venom subsequently, and 2 pulses of 100 mM GABA were applied to test the activity after the venom incubation. For K_V_1.3 channels, cells were clamped at a holding potential of −90 mV. The K^+^ current was evoked by 500 ms depolarizations to 0 mV, followed by 500 ms pulses to −50 mV. Finally, for Na_V_1.2 and Na_V_1.4 channels, cells were clamped at −90 mV holding potential and the Na^+^ current was evoked by 100 ms depolarizations to 0 mV. Lyophilized venoms were resuspended in ND96 solution for the final concentration of 0.5 to 1 mg/mL. For all experiments with ion channels, the amount of ~10 µg of crude venom was applied by pipetting directly on the bath, without flow. For the α1β1δε nAChR and α7 nAChR, the venoms were incubated for ~50 s, and for α1β2 GABA_A_R, ~1 min. The bath perfusion of ND96 solution to wash the oocytes was under the gravitational flow (~2 mL/min).

## Figures and Tables

**Figure 1 ijms-23-07714-f001:**
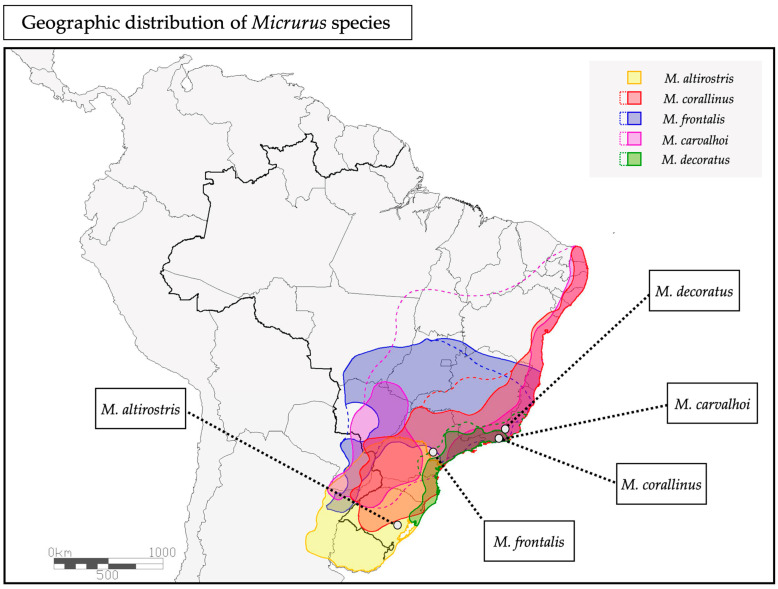
Geographic distribution of the five *Micrurus* species. The colored areas represent the current consolidated geographic distribution of the *Micrurus* species, and the dotted colored lines represent the distribution areas under scientific revision. The areas where the specimens used in this work were collected are indicated in the map with the names of the species.

**Figure 2 ijms-23-07714-f002:**
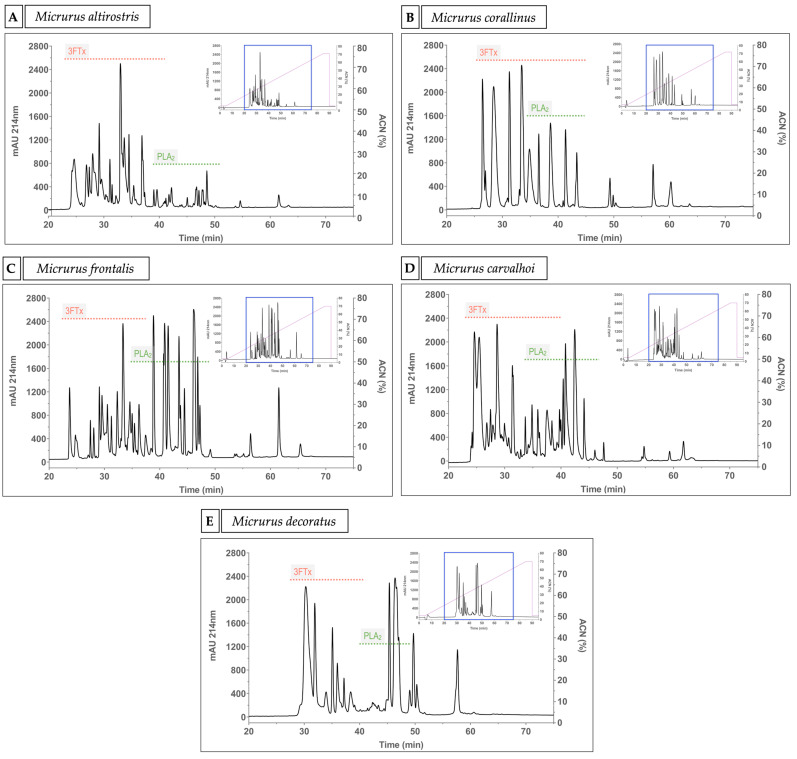
Crude venom chromatographic profiles. (**A**) *Micrurus altirostris*, (**B**) *Micrurus corallinus*, (**C**) *Micrurus frontalis*, (**D**) *Micrurus carvalhoi* and (**E**) *Micrurus decoratus* venoms. A total of 500 µg of crude venom was used to obtain the RP-HPLC profile. Based on the published proteome information of *Micrurus* species venoms [8,9,10], we show in the figure the approximate elution regions of 3FTxs and PLA_2_s, marked by dotted lines in red and green, respectively. The right superior panel in each figure shows the full area of the chromatogram, where the blue box represents the zoomed area and the purple line represents the acetonitrile (ACN) gradient.

**Figure 3 ijms-23-07714-f003:**
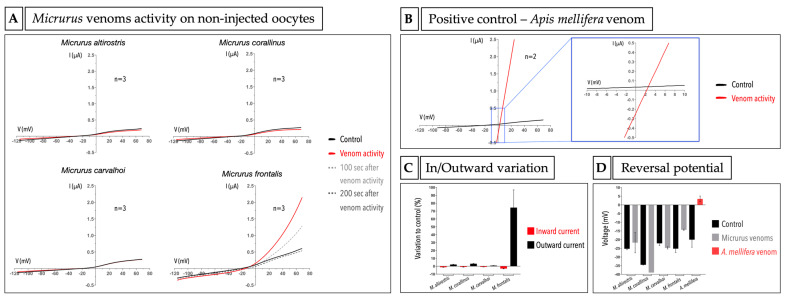
Cytotoxic evaluation of crude venoms on *X. laevis* non-injected oocytes. (**A**) *Micrurus altirostris*, *Micrurus corallinus*, *Micrurus carvalhoi* and *Micrurus frontalis* venom activity. (**B**) Positive control-*Apis mellifera* venom activity. The black lines show the control, and the red lines show the venom activity. In panel (**A**), the dotted dark gray and light gray lines show 100 s and 200 s of past time from the venom activity, respectively. The blue box shows a zoom of the graph on panel (**B**). The graphs show the trace of one experiment from a series of independent experiments. (**C**) Inward and outward current graph. (**D**) Reversal potential graph.

**Figure 4 ijms-23-07714-f004:**
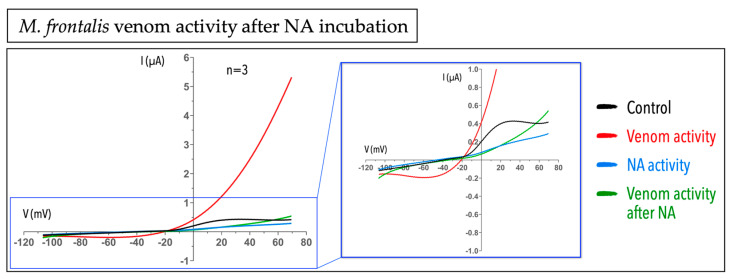
*Micrurus frontalis* venom causes a Cl^−^ outward current by acting directly or indirectly on Ca^2+^-activated Cl^−^ channels in *X. laevis* oocytes. The black line shows the control (the oocyte perfused with ND96); the red line shows the activity of 50 µg of *M. frontalis* venom on oocytes; the blue line shows the oocyte current in the presence of 100 µM of NA; and the green line shows the activity of 50 µg of *M. frontalis* venom after cell incubation with 100 µM of NA, where the outward current evoked by the venom is no longer observed. The blue box on the right of the panel shows a zoom of the graph. The graphs show the trace of one experiment from a series of independent experiments.

**Figure 5 ijms-23-07714-f005:**
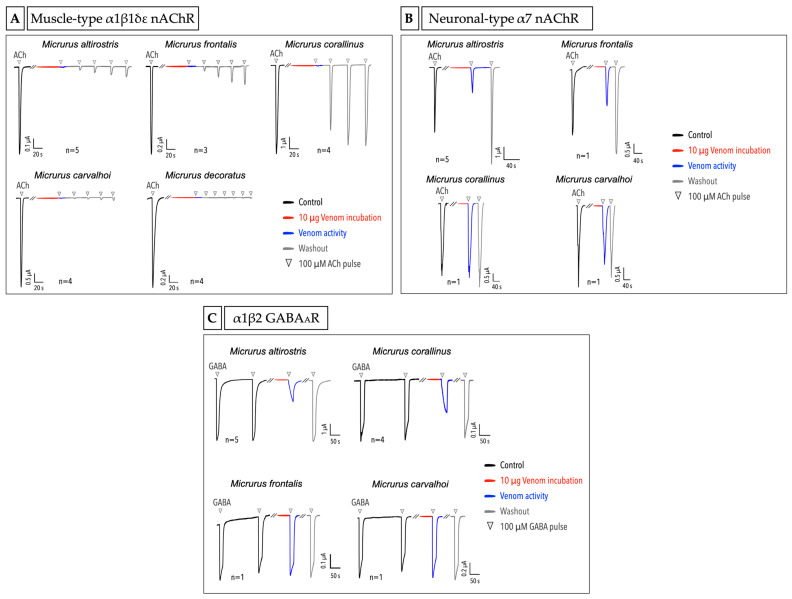
Screening of crude venoms from different *Micrurus* species in different ligand-gated ion channels. (**A**) Muscle-type α1β1δε nAChR, (**B**) Neuronal-type α7 nAChR and (**C**) α1β2 GABA_A_R. The black lines represent the control pulses, the red lines represent the venom incubation, the blue lines represent the receptor activity after the incubation with 10 µg of crude venom, and the gray lines represent the agonist pulse after the cell washout. The triangles mark the point where the application of the agonist was started. The *n* from each experiment is shown as well as the current/time reference bar. The graphs show the trace of one experiment from a series of independent experiments, with the exception of tests with *n* = 1.

**Figure 6 ijms-23-07714-f006:**
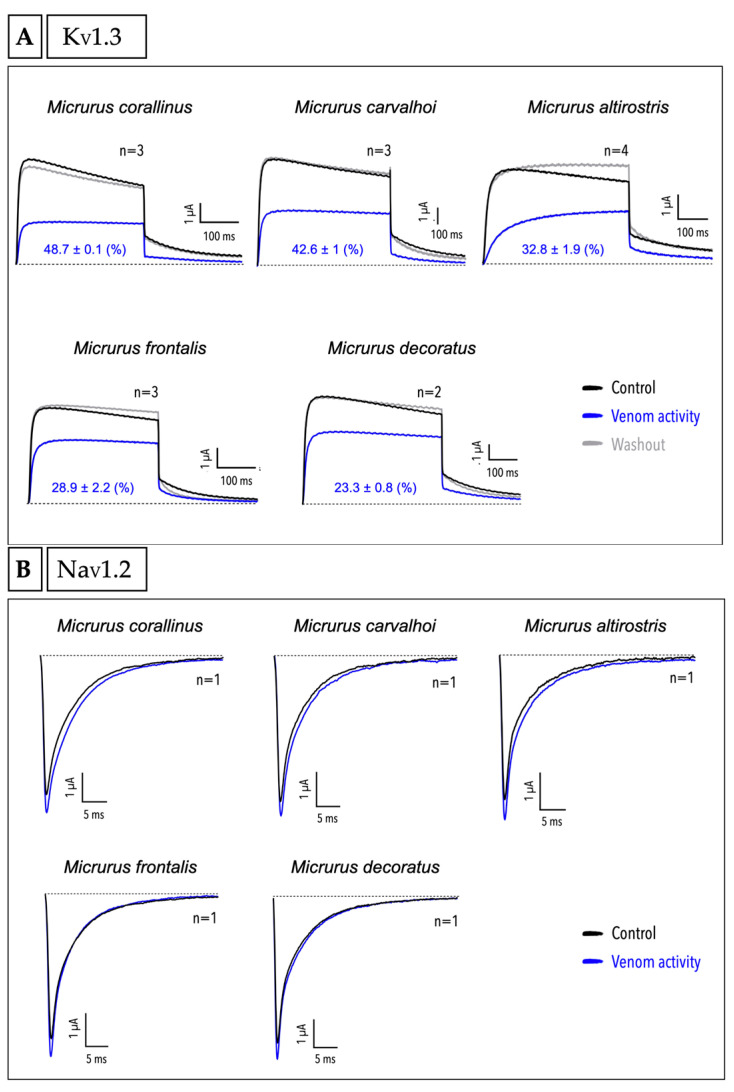
Screening of crude venoms from different *Micrurus* species in different voltage-gated ion channels. (**A**) K_V_1.3. The inhibition percentage and the SEM are shown in blue for each graph. (**B**) Na_V_1.2. The black lines represent the control and the blue lines represent the channel response after incubation with 10 µg of crude venom. The dotted lines represent the zero current level. The *n* from each experiment is shown as well as the current/time reference bar. The graphs show the trace of one experiment from a series of independent experiments, with the exception of tests with *n* = 1.

## Data Availability

Not applicable.

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
