# Peer review of "Pharmacological Screening of Venoms from Five Brazilian *Micrurus* Species on Different Ion Channels"

_ijms, 2022, doi:10.3390/ijms23147714_

Round 1
Reviewer 1 Report
Interesting review, and well presented. I have a very few comments to make.
1. Page 1, line 40: I think there is ambiguity regarding North American, or South American or both?
2. Figure 3: There are no data points on the IV plots, nor error bars. The change in reversal potential bar plot may presumably mean there is a degree of K+ selectivity to the leaky membrane (?). Can you please explain more about what is supposed to be going on, including in the figure legend.
3. Figure 4: The authors present strange 'fitted' curves with no datapoints, or indication of error. Can this be put right? At the moment I have no compelling reason to believe these curves.
4. Figure 6: Voltage-clamp protocols are not shown. There are no details of the effect of the toxins tested on the activation kinetics or activation voltage-dependence of the expressed Na+ channels, only on inactivation.
5. Page 8, line 262 and 263: 'cytotoxic' due to molecules that 'can cause cell death' . This appears to be a tautology, saying the same thing twice using different words. Please modify the sentence to avoid this.
Author Response
Page 1, line 40: I think there is ambiguity regarding North American, or South American or both?
A: We are sorry, but we did not understand what you meant by ‘ambiguity’. This information is regarding the whole American continent. Specifically, the south of the continent and the north of the continent are pointed out in the article.
Figure 3: There are no data points on the IV plots, nor error bars. The change in reversal potential bar plot may presumably mean there is a degree of K+ selectivity to the leaky membrane (?). Can you please explain more about what is supposed to be going on, including in the figure legend.
A: In the electrophysiology experiments, we represent a set of data (e.g., n=3) showing just one representative graph. To avoid any confusion, we will indicate in the legend that the trace on the figure is a representative graph of one of the experiments. So, for example: In a set of 3 independent experiments, we selected one representative graph to show in the figure.
Regarding the reversal potential, we explained in the ‘Discussion section’ the following: “In the electrophysiology experiments using the voltage ramp protocol, when a given venom is forming pores that are non-selective for ions and their conduction, typically two phenomena are observed: 1) a shift of the reversal potential to(wards) 0 mV, and 2) a significant increase in the chord conductance.”
Since the cytotoxic effect can cause a pore (a hole) in the membrane, as mentioned above, it is not selective for any ion, it is rather non-selective.
Figure 4: The authors present strange 'fitted' curves with no datapoints, or indication of error. Can this be put right? At the moment I have no compelling reason to believe these curves.
A: In the same way shown in figure 3, in figure 4 the IV plot (current (I) versus voltage (V)), we did not fit a curve, it is the data itself. We export the data obtained using a voltage ramp protocol (a pre-defined protocol that changes the voltage continuously and automatically during the data acquisition) from the electrophysiology setup, that generates a trace which was represented as a figure made in the Prisma software.
Figure 6: Voltage-clamp protocols are not shown. There are no details of the effect of the toxins tested on the activation kinetics or activation voltage-dependence of the expressed Na+ channels, only on inactivation.
A: All the protocols are detailed in the ‘Methods section’ . For figure 6 specifically, they are described in section 5.3.3, in the second paragraph. The tests on Nav1.2 and 1.4 showed that the crude venoms have no effect on these channels. As so, there is no activation/inactivation to discuss.
Page 8, line 262 and 263: 'cytotoxic' due to molecules that 'can cause cell death' . This appears to be a tautology, saying the same thing twice using different words. Please modify the sentence to avoid this.
A: We agreed with the referee and we simply removed the sentence "due to the presence of molecules that can cause cell death".
Reviewer 2 Report
This is a beautiful piece of neurophysiology work and a valuable contribution to the literature.
I have only two comment that the authors can consider as optional:
1. Specialists reading this manuscript will generally know if you are talking about 3FTX vs. sPLA2 toxins. Most readers will not immediately associate the various channels/AChRs with 3FTX. To that end, it might make the paper more accessible to readers if you identify somewhere early on that the channels/recepters to which you are referring most often I the paper are pre- or post-synaptic and the associated class of toxins.
2. In figure 2, none of the most interesting peaks are labeled. For example, it would be nice to see which main peak/s are 3FTX vs. sPLA2 so that readers can compare the peaks to the text (e.g. M. frontalis being 49% sPLA2/42%3FTx compared to other that 3FTX is ~80%).
This is a work worthy of congratulations. Thank you for sharing this technical elegant and clinically relevant body of work.
Author Response
Specialists reading this manuscript will generally know if you are talking about 3FTX vs. sPLA2 toxins. Most readers will not immediately associate the various channels/AChRs with 3FTX. To that end, it might make the paper more accessible to readers if you identify somewhere early on that the channels/receptors to which you are referring most often I the paper are pre- or post-synaptic and the associated class of toxins.
A: We agreed with this suggestion, and we included a sentence in the discussion section lines 335-338. “Despite the wide range of biological targets, classically, the main 3FTx activity is the disturbance of the cholinergic system, specifically the inhibition of the muscarinic or nicotinic AChRs [31]”.
In figure 2, none of the most interesting peaks are labeled. For example, it would be nice to see which main peak/s are 3FTX vs. sPLA2 so that readers can compare the peaks to the text (e.g. M. frontalis being 49% sPLA2/42%3FTx compared to other that 3FTX is ~80%).
A: We agree with your suggestion but, at this point, we did not characterize the proteomes of the venoms used in this work. Even though, since there are available in the literature the proteomes for 4 out of the 5 venoms used in this work, we changed figure 2 and indicated approximately the regions of the two main classes of toxins – 3FTx and PLAs, as you suggested.
This is a work worthy of congratulations. Thank you for sharing this technical elegant and clinically relevant body of work.
A: We thank the referee for the compliment and the suggestions that improved our manuscript.